# Asymmetric Dual-Lens Video Deblurring

**Zeyu Xiao      Xinchao Wang**[*]
National University of Singapore
zeyuxiao@nus.edu.sg,    xinchao@nus.edu.sg

## Abstract

Modern smartphones often feature asymmetric dual-lens systems, capturing wide-angle and ultra-wide views with complementary perspectives and details. Motion and shake can blur the wide lens, while the ultra-wide lens, despite lower resolution, retains sharper details. This natural complementarity offers valuable cues for video deblurring. However, existing methods focus mainly on single-camera inputs or symmetric stereo pairs, neglecting the cross-lens redundancy in mobile dual-camera systems. In this paper, we propose a practical video deblurring method, AsLeD-Net, which recurrently aligns and propagates temporal reference features from ultra-wide views fused with features extracted from wide-angle blurry frames. AsLeD-Net consists of two key modules: the adaptive local matching (ALM) module, which refines blurry features using $K$-nearest neighbor reference features, and the difference compensation (DC) module, which ensures spatial consistency and reduces misalignment. Additionally, AsLeD-Net uses the reference-guided motion compensation (RMC) module for temporal alignment, further improving frame-to-frame consistency in the deblurring process. We validate the effectiveness of AsLeD-Net through extensive experiments, benchmarking it against potential solutions for asymmetric lens deblurring.

## 1 Introduction

Modern smartphones increasingly feature asymmetric dual-lens systems, combining wide-angle and ultra-wide lenses to capture scenes with complementary perspectives and details (see Figure 1). However, during dynamic scenes or camera shake, focal length and aperture differences introduce blur discrepancies. For example, on the iPhone 14 Plus, ultra-wide lenses typically have shorter focal lengths (*e.g.*, 13mm) and smaller apertures (*e.g.*, f/2.4) compared to wide-angle lenses (*e.g.*, 26mm, f/1.5). The shorter focal length of ultra-wide lenses results in a deeper depth of field, reducing defocus blur but making them more prone to motion blur due to the longer exposure time required for sufficient light intake. In contrast, wide-angle

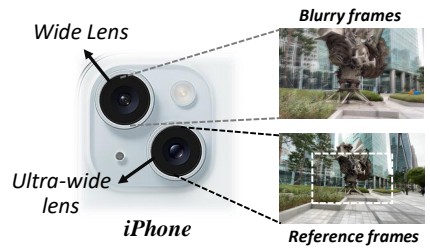

Figure 1: A typical dual-lens camera system on iPhone in practice.

lenses with a larger aperture capture more light, allowing for faster shutter speeds and reducing motion blur while maintaining a shallower depth of field that enhances subject-background separation. These complementary characteristics provide valuable cues for video deblurring, as shown in Figure 2.

Existing video deblurring methods primarily focus on single-camera inputs or symmetric stereo pairs. For instance, flow-guided bidirectional propagation methods [8, 21, 40] align features using deformable convolutions and attention mechanisms, while VRT and RVRT [37, 38] employ spatio-temporal self-attention to aggregate information across video frames. However, these methods do not leverage the cross-lens redundancy inherent in mobile dual-camera systems. Although several

---

[*]Corresponding Author

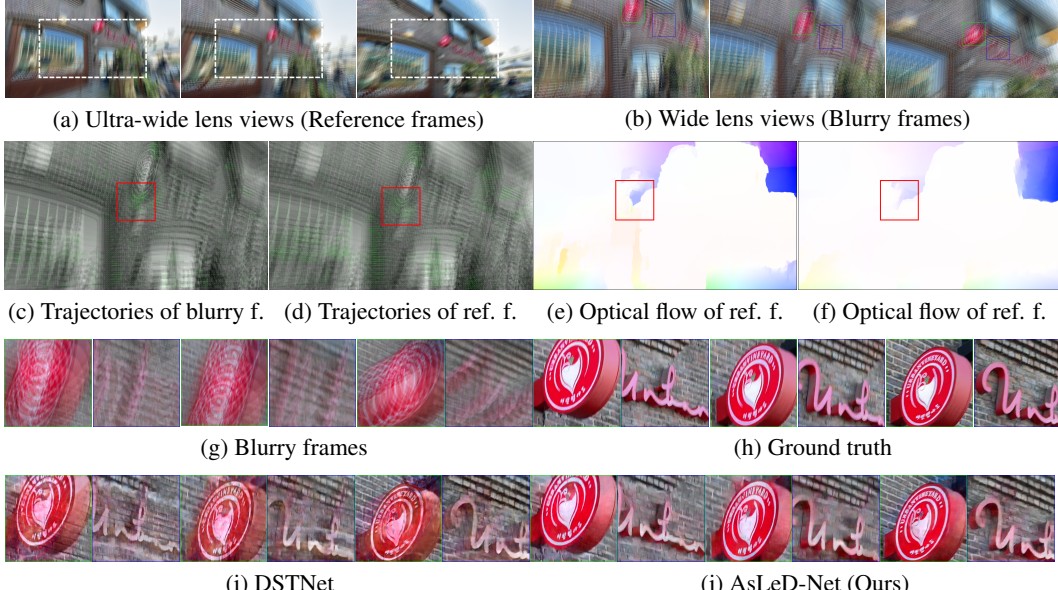

(a) Ultra-wide lens views (Reference frames)      (b) Wide lens views (Blurry frames)

(c) Trajectories of blurry f.   (d) Trajectories of ref. f.   (e) Optical flow of ref. f.   (f) Optical flow of ref. f.

(g) Blurry frames      (h) Ground truth

(i) DSTNet      (j) AsLeD-Net (Ours)

Figure 2: Examples of asymmetric dual-lens video deblurring. (a) Reference frames captured using the ultra-wide lens. (b) Blurry frames captured using the wide-angle lens. (c-d) Motion discrepancy visualization through trajectory comparisons (Please zoom in and best viewed on the screen). (e-f) Optical flow visualization of blurry and reference frames. (g) Input blurry sequences. (h) Ground truth. (i) Results from a state-of-the-art method (DSTNet) [52]. (j) Our AsLeD-Net results. In this figure, "f." is short for frames, and "ref." is short for reference, respectively. We propose the *first* deep learning method for *asymmetric dual-lens video deblurring*, where ultra-wide lens videos (shorter focal length) provide motion priors to restore clear wide-lens blurry sequences. This is motivated by the observation that ultra-wide lenses inherently suffer *less motion blur* due to their focal properties: shorter focal lengths reduce perceived motion parallax, enabling more stable scene capture despite the lower resolution. AsLeD-Net delivers high deblurring accuracy with strong temporal coherence.

image restoration techniques have been developed for dual-camera setups, including dual-lens super-resolution [70, 94, 31, 84, 28, 69, 7, 76, 85, 95, 53] and dual-lens stereo matching [12, 64, 90], as well as reference-based image super-resolution [82, 62, 44, 6, 93, 56, 23, 24, 20, 29, 92, 25], video deblurring in an *asymmetric dual-lens* setting remains unexplored. In this paper, we introduce Asymmetric dual-Lens video Deblurring (AsLeD), which exploits the complementary information between blurry and reference videos captured from asymmetric dual-camera systems. Unlike traditional video deblurring approaches, AsLeD explicitly considers the structural disparities between the two views to enhance the restoration process.

AsLeD effectively models the relationship between blurry and reference frames in asymmetric dual-lens settings. At each time step, the two frames share nearly identical content within their overlapping field of view (FoV) (top and middle rows of the leftmost column in Figure 2). As the video progresses, neighboring reference frames provide sharp details that help recover regions beyond the overlapped FoV (bottom row of the leftmost column in Figure 2). By leveraging these cross-frame correspondences, AsLeD enhances video restoration by utilizing multi-frame redundancy in asymmetric multi-camera setups.

AsLeD-Net employs a dual-branch architecture built upon the BasicVSR framework to exploit asymmetric dual-lens characteristics. The base branch uses a bidirectional recurrent design [8, 21] for temporal feature propagation, while the reference branch processes ultra-wide lens inputs with reduced motion blur. This asymmetric design incorporates three key modules in the reference branch: (1) The adaptive local matching (ALM) module: Establishes semantic-aware correspondences between blurry frames (base branch) and clear reference frames (ultra-wide branch) using $K$-nearest neighbor feature aggregation, effectively transferring structural details. (2) The difference compensation (DC) module: Bridges feature discrepancies between the branches, preserving edge sharpness in fast-moving regions and enhancing spatial-temporal consistency. (3) The reference-guided motion compensation (RMC) module: Aligns features across time steps using optical flow guided by ultra-wide reference frames, eliminating cumulative errors in traditional flow-based warping. The base branch progressively

fuses ALM-refined features, DC-compensated residuals, and RMC-aligned references via attention-guided fusion blocks. This hierarchical integration enables the model to leverage the base branch's reconstruction capabilities and the reference branch's blur-insensitive motion priors. As shown in Figure 2, AsLeD-Net outperforms DSTNet [52], enhancing texture details like the text on the wall and the coffee shop logo while preserving strong temporal coherence.

The contributions of this work are summarized as follows: (1) We formalize asymmetric dual-lens video deblurring (AsLeD), leveraging cross-lens redundancy from a sharper ultra-wide reference to assist a blurry wide view, extending prior dual-lens *image* deblurring to the video setting. (2) We present AsLeD-Net with three task-specific modules: ALM for structure-aware K-nearest-neighbor reference aggregation, DC for cross-view spatial consistency, and RMC for temporal alignment. (3) AsLeD-Net achieves superior quantitative and qualitative results on AsLeD.

## 2    Related Works

**Video deblurring.** Unlike image deblurring [48, 67, 87, 79, 77, 35], video deblurring benefits from spatio-temporal cues to enhance restoration quality [51, 33, 99, 98, 88, 72, 66, 80, 10, 21, 22, 49, 65, 81, 37, 58, 89]. Deep learning-based methods [3, 2, 97, 96, 54, 55, 36, 78] have become the dominant approach. Recurrent video deblurring models exploit temporal dependencies for progressive feature propagation [98, 91, 72, 10, 39, 49, 52, 32]. For example, STRCNN [27], RDN [74], and IFRNN [49] adopt recurrent architectures for sequential feature refinement. STFAN [100] employs dynamic filters, while PVDNet [63] integrates a blur-invariant flow estimator. Recent methods leverage bidirectional propagation for improved restoration [8, 101, 21, 40, 88]. BasicVSR++ [8] introduces aggressive bidirectional propagation, while RNN-MBP [101] incorporates multi-scale bidirectional updates. However, error accumulation remains challenging for long-range temporal modeling [21]. Spatio-temporal transformers enhance video deblurring by capturing long-range dependencies [37, 40, 39, 37, 38]. Recently, Zhang *et al.* [89] propose a spatio-temporal sparse Transformer for efficient video deblurring.

**Dual-lens image restoration.** Asymmetric dual-lens systems, commonly found in smartphones, consist of an ultra-wide lens and a wide lens with different focal lengths. These systems capture the same scene with varying FoVs. Typically, the ultra-wide lens (short focal length, large FoV) is the main lens, while the wide lens (longer focal length, narrower FoV) provides higher resolution within the overlapped FoV. This configuration enables various vision tasks by leveraging the complementary imaging capabilities of the two lenses. Previous works have explored dual-lens systems for image refocusing [1], correspondence estimation [12, 64, 90] and image super-resolution [70, 94, 31, 84, 28, 69, 7, 76, 85, 95, 53], where telephoto images serve as high-resolution references to enhance wide-angle images. Beyond image super-resolution, other tasks such as image deblurring [46, 61, 59], novel view synthesis [75, 83, 45], high-dynamic-range imaging [34, 60], and image colorization [15] have also been investigated, demonstrating the practical benefits of cross-lens redundancy. Among prior studies, the most similar to ours are Mohan *et al.* [46], who deblur static dual-lens image pairs by enforcing cross-view and depth consistency in unconstrained capture settings. Lai *et al.* [30] deblur face regions in still photos using a synchronized sharp ultra-wide reference. We extend this line from images to video by integrating reference-guided cross-view alignment with recurrent temporal propagation to preserve frame-to-frame coherence.

**Reference-based image restoration.** Another related topic is reference-based image restoration, which includes tasks such as reference-based image super-resolution [82, 62, 44, 6, 93, 56, 23, 24, 20, 29, 92, 25], image deblurring [41, 42] and burst image restoration [14, 19, 4, 26, 16, 5, 17, 73, 18, 43, 29]. These tasks leverage auxiliary high-quality images to enhance the restoration of a degraded input. In this work, we address reference-based video deblurring in the context of asymmetric dual-lens smartphone cameras. Unlike existing approaches focusing on still images, our method utilizes cross-lens redundancy in videos to achieve high-quality deblurring, effectively integrating information from both lenses.

## 3    Method

### 3.1    Overview

The proposed AsLeD-Net reconstructs a high-quality video $\hat{\mathcal{I}} \in \mathbb{R}^{T \times H \times W \times 3}$ from a blurry input $\mathcal{I}^B \in \mathbb{R}^{T \times H \times W \times 3}$ captured with a wide lens, leveraging a relatively sharper reference video

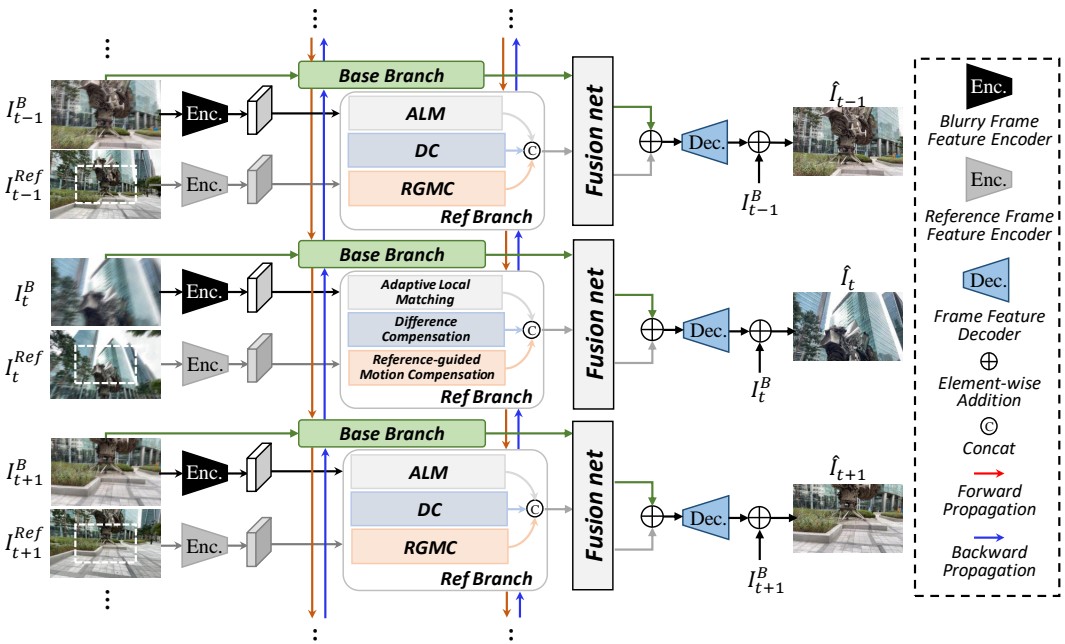

Figure 3: Overview of the proposed AsLeD-Net. AsLeD-Net processes blurry video frames from a wide lens and reference frames from an ultra-wide lens using a dual-branch architecture. The base branch reconstructs the blurry frames with bidirectional frame propagation [9], while the reference branch extracts asymmetric information from the ultra-wide frames. Features from both branches are encoded separately and then refined by ALM, DC, and RMC modules. The concatenated features are merged through residual blocks and fed to the frame decoder to generate the final clear output.

$\mathcal{I}^{Ref} \in \mathbb{R}^{T \times H \times W \times 3}$ taken simultaneously by an ultra-wide len in a dual-lens camera (*e.g.*, iPhone). The objective is to restore $\hat{\mathcal{I}}$ to closely match the ground truth $\mathcal{I}^{GT}$, effectively exploiting the complementary information from the reference view. Here, $T$, $H$, and $W$ denote the number of frames, height, and width, respectively. Following previous video deblurring and restoration methods [8, 9], we adopt a bi-directional recurrent architecture for its simplicity and efficiency. The overall framework of our proposed AsLeD-Net is shown in Figure 3.

The base branch, identical to BasicVSR, reconstructs blurry videos $\mathcal{I}^B$ captured with a wide lens, using bidirectional frame propagation based on optical flow to extract features $\boldsymbol{F}^{Base}$. To exploit the unique properties of the AsLeD task, we introduce a second reference branch, which processes ultra-wide lens inputs with reduced motion blur. Since reference and blurry frames have different FoV and contain varying information, we center-crop and resize the reference frames to match the blurry frame size. We leverage the reference branch to extract asymmetric reference information despite potential colour discrepancies due to lens differences.

For the blurry frame $I_t^B$ and its corresponding reference frame $I_t^{Ref}$ at time $t$, we first feed them to separate feature encoders with non-shared weights to obtain feature representations $F_t^B$ and $F_t^{Ref}$, each consisting of $N_1$ residual blocks. These features are then processed by the ALM, DC, and RMC modules, which transfer structural details from the reference frames, bridge feature discrepancies while preserving edge sharpness in fast-moving regions, and align features across time steps using optical flow guided by the ultra-wide reference frames. After concatenation, the outputs $F_t^{ALM}$, $F_t^{DC}$, and $F_t^{RMC}$ are merged with the base branch's output $F_t^{Base}$ by feeding them to residual blocks, resulting in $F_t$. Finally, $F_t$ is input into a frame decoder, composed of $N_2$ residual blocks, to generate the final reconstructed result $\hat{I}_t$, with a residual connection from the input added.

Our approach does not assume perfect geometric calibration or strict cross-view alignment. Instead, AsLeD-Net learns content-aware feature alignment that tolerates cross-lens baselines and photometric shifts, enabling reference guidance under realistic dual-lens configurations. In practice, we retain cross-view geometric and photometric discrepancies and rely on content-aware, data-driven feature alignment, while ensuring frame-level temporal synchronization.

## 3.2 Adaptive Local Matching

Motion blur erases high-frequency details and disrupts spatial coherence, leading to texture ambiguity and structural distortions. Recovering these details is particularly challenging due to the ill-posed nature of the problem. To address this issue, we exploit the complementary information available in reference frames, which are captured under similar conditions but with less or no blur. These frames retain rich structural cues that can guide the restoration of the blurry frame, providing a more robust mechanism for handling local details. The core idea is to adaptively leverage the reference frame's features based on local context, enabling more accurate restoration of the blurred image.

Without loss of generality, we focus on the time step $t$ to illustrate the ALM module. Given the extracted blurry frame feature $F_t^B \in \mathbb{R}^{C \times H \times W}$ at timestamp $t$, the ALM module aims to refine these blurry local features by incorporating the complementary reference feature $F_t^{Ref} \in \mathbb{R}^{C \times H \times W}$ captured under similar conditions but with less blur. These deep features can be represented as a set of $H \times W$ local representations, each of size $C$-dimensional. To match the blurry frame with the reference features, we first compute a similarity map $S_t$ to measure the cosine similarity between each blurry local feature $F_t^{B(i)}$ and each reference feature $F_t^{Ref(j)}$

$$S_t^{(i,j)} = \cos\left(F_t^{B(i)}, F_t^{Ref(j)}\right) = \frac{F_t^{B(i)\top} F_t^{Ref(j)}}{\|F_t^{B(i)}\| \cdot \|F_t^{Ref(j)}\|}, \tag{1}$$

where $i \in \{1, \ldots, N_t^B\}$ and $j \in \{1, \ldots, N_t^{Ref}\}$, and $N_t^B$ and $N_t^{Ref}$ represent the number of local representations in the blurry and reference frames, respectively. $S_t^{(i,j)}$ represents the similarity between the $i$-th blurry local representation and the $j$-th reference local representation.

For each blurry local representation, we select its $K$-nearest neighbors from the reference features and fuse these local representations into a reference representation $\phi_{t,ref}^{(i)}$

$$\phi_{t,ref}^{(i)} = \sum_{k=1}^{K} \alpha_t^{(k)} \times F_t^{Ref(k)}, \tag{2}$$

where $\sum_{k=1}^{K} \alpha_t^{(k)} = 1$, $\alpha_t^{(k)} > 0$, and $\alpha_t^{(k)}$ is obtained using the softmax function. The reference representations then adaptively refine the corresponding blurry features. Finally, the blurry and reference representations are concatenated and processed by an adaptive layer $g_\theta$ to obtain the fused local representations

$$F_t^{ALM} = g_\theta([(\phi_{t,ref}, F_t^B)]), \tag{3}$$

where $\phi_{t,fuse}$ represents the fused local representations, and $g_\theta$ denotes the adaptive fusion layer. The cosine similarities in Eq. (1) form a query-reference affinity used to rank reference tokens; the top-$K$ are aggregated with softmax-normalized weights. The adaptive layer $g_\theta$ in Eq. (3) is a lightweight $1 \times 1$ convolution applied to the concatenated $[\phi_{t,\text{ref}}, F_t^B]$, enabling content-aware fusion with low overhead. This process facilitates information fusion by incorporating relevant reference features to enhance the blurry frame representation.

## 3.3 Difference Compensation

In AsLeD, while blurry and reference images share structural consistency, focus, exposure, and sensor processing differences create subtle feature mismatches, making direct concatenation suboptimal. The DC module provides a

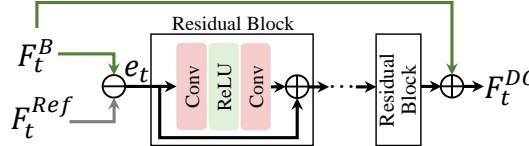

Figure 4: Overview of the difference compensation (DC) module.

simple yet effective solution by adaptively modeling residual information. It ensures that only beneficial details are aggregated while preserving spatial consistency. Unlike complex alignment-based approaches, the DC module efficiently refines blurry features with relevant reference information, preventing artifacts and enhancing deblurring with minimal computational overhead.

Given the blurry frame feature $F_t^B$ and its corresponding reference frame feature $F_t^{Ref}$ at time $t$, the goal of the DC module is to refine the blurry features by effectively leveraging reference information. As shown in Figure 4, we predict the difference between the blurry feature $F_t^B$ and the reference

feature $F_t^{Ref}$, enhancing the residual information $e_t$ to facilitate effective feature aggregation and adaptive refinement. The difference compensation between $F_t^{Ref}$ and $F_t^{Ref}$ can be summarized as

$$e_t = F_t^B - F_t^{Ref}, \quad \hat{e}_t = f(e_t), \quad F_t^{DC} = F_t^B + \hat{e}_t, \tag{4}$$

where $e_t$ represents the residual feature that captures useful differences between the blurry and reference features, and $f(\cdot)$ denotes the cascaded residual blocks. This progressive refinement adaptively integrates reference information while preserving spatial consistency, ensuring effective feature aggregation for improved deblurring.

### 3.4 Reference-Guided Motion Compensation

In the base branch of AsLeD-Net, optical flow is computed between the current and previous blurry frames for warping. While this approach has its merits, it often introduces limitations that degrade deblurring performance. Specifically, both the current and previous blurry frames may contain significant motion blur and noise, leading to inaccuracies in optical

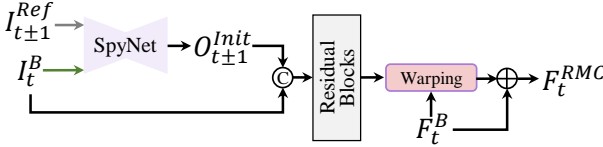

Figure 5: Overview of the reference-guided motion compensation (RMC) module.

flow estimation. These imperfections can cause misalignment during warping, resulting in artifacts or an inability to capture fine motion details, ultimately reducing the quality of deblurring.

To address these issues, we propose the RMC module, which estimates optical flow between the current blurry frame and a sharper reference frame (previous or next) rather than between consecutive blurry frames. It leverages two complementary motions: temporal flow within the base view and cross view flow guided by the sharper reference; the latter accounts for differences in field of view, perspective, and resolution, is more reliable under heavy blur, and reduces misalignment to improve motion tracking. As is shown in Figure 5, given the current blurry frame $I_t^B$ and the previous reference frame $I_{t-1}^{Ref}$, we first use SpyNet [57] to estimate the initial optical flow $O_{t-1}^{Init}$ between these two frames. However, this initial flow estimate may not be sufficiently accurate due to factors such as noise, motion blur, or misalignment, which can affect the precision of the optical flow calculation. To improve the flow estimate, we concatenate the current blurry frame $I_t^B$ and the initial flow estimate $O_{t-1}^{Init}$, then feed this concatenated pair to residual blocks to refine the flow. This refined flow is expected to capture fine details and correct the misalignment introduced by the initial estimation

$$O_{t-1}^{Init} = \text{SpyNet}(I_t^B, I_{t-1}^{Ref}), \quad O_{t-1}^{Refined} = f\left([I_t^B, O_{t-1}^{Init}]\right). \tag{5}$$

The refined flow is not directly supervised; it is learned implicitly via end-to-end reconstruction loss, in line with prior video restoration designs where flow acts as an auxiliary representation to facilitate motion compensation rather than a target itself.

After refining the optical flow, we perform warping using the refined flow and the current blurry frame $I_t^B$. The result is then combined with the residual information to obtain the motion-compensated feature $F_t^{RMC}$, which is more accurate in capturing the motion and structural details of the scene. This process ensures that the reference frame's motion is effectively transferred to the blurry frame, improving alignment and reducing artifacts. The above process can be formalized as

$$F_t^{RMC} = \text{warp}(I_t^B, O_{t-1}^{Refined}) + f(I_t^B). \tag{6}$$

By refining the optical flow and leveraging the residual information, the RMC module ensures a more precise alignment between the current blurry frame and the reference frame, leading to improved deblurring performance.

## 4 Experiments

### 4.1 Experimental Settings

**Datasets.** We use the RealMCVSR dataset [31] for our experiments. Originally designed for multi-view video super-resolution, RealMCVSR consists of triplets captured with ultra-wide, wide-angle, and telephoto lenses. In this setup, the wide-angle and telephoto videos share the same spatial dimensions as the ultra-wide video but have $\times 2$ and $\times 4$ higher resolutions, respectively. For the

AsLeD task, we adapt the dataset by using the ultra-wide video as the reference input and the wide-angle video as the blurry input. Since the wide-angle video has twice the resolution of the ultra-wide video, we center-crop and upsample the ultra-wide frames to match its resolution. We follow the original data split of the RealMCVSR dataset. To simulate motion blur, we generate blurred frames by the widely used technique of averaging multiple consecutive frames of the video captured by different lenses [65, 48, 47, 71, 50]. We generate blurry frames for training by averaging every 7 consecutive frames, simulating varying motion blur intensities. We synthesize motion blur by averaging seven consecutive frames for both training and testing, a widely used approximation that does not fully model real-world blur. For real captures, we use an iPhone 14 Plus whose two lenses introduce a physical baseline and distinct imaging characteristics, yielding cross-view shifts and photometric differences. We intentionally avoid spatial or color pre-alignment and ensure frame-level temporal synchronization with the DoubleTake APP. Note that RealMCVSR is captured on iPhone 12 Pro Max, whereas our real tests use iPhone 14 Plus, introducing cross-device differences in optics and ISP to assess generalization. Evaluations on these real videos demonstrate robust generalization beyond the averaging assumption.

**Implementation details.** We use 7 frames as input during training, with a mini-batch size of 4 and an input frame resolution of $128 \times 128$. We apply data augmentation techniques to the training data, including horizontal flips and random rotations of $90°$, $180°$, and $270°$. AsLeD-Net is trained for 300K iterations using the Adam optimizer with a Cosine Annealing learning rate scheduler. Network architecture parameters are set to $N_1 = 1$ and $N_2 = 30$. The number of channels is 64, and $K$ in ALM is set to 3. Supervision is enforced using the Charbonnier loss [71] via $\mathcal{L} = \sqrt{\|\hat{\mathcal{I}} - \mathcal{I}^{GT}\|^2 + \varepsilon^2}$, where $\varepsilon$ is set to $1 \times 10^{-3}$ in our experiments. We omit the subscript $t$ for simplicity. The initial learning rate for AsLeD-Net is $1 \times 10^{-4}$. Training is conducted on an NVIDIA RTX 3090 GPU.

**Inference settings.** We evaluate the reconstructed results using PSNR, SSIM, and LPIPS on the RGB channels to assess fidelity and perceptual quality.

## 4.2 Quantitative and Qualitative Comparisons

We compare the proposed AsLeD-Net against a diverse set of baseline methods to evaluate its effectiveness for the AsLeD task. These baselines encompass a wide range of potential approaches, aiming to cover as many varied and rich methodologies as possible: (1) Single-image deblurring and restoration methods: including MIMOUNet [13], MIMOUNet++ [13], NAFNet [11], and Restormer [86]. (2) Video deblurring and restoration methods: including IFIRNN [49], DBN [65], EDVR [71], BasicVSR [9], and BasicVSR++ [8]. (3) Reference-based video restoration method: specifically RefVSR [31]. Notably, some of these methods are not initially designed for deblurring. For approaches primarily intended for super-resolution tasks, we remove the upsampling operation and instead apply a

Table 1: Quantitative evaluation on the RealMCVSR testset. We mark the best and the second best results in **bold** and underline, respectively. #Params means the number of network parameters (M). Time costs (ms) are measured on blurred frames with a resolution of $256 \times 256$ using an NVIDIA GTX 1080 Ti GPU.

| Method | RealMCVSR | | | Costs | |
|---|---|---|---|---|---|
| | PSNR↑ | SSIM↑ | LPIPS↓ | #Params | Time |
| Blur frame | 18.22 | 0.8009 | 0.4319 | - | - |
| MIMOUNet | 24.50 | 0.8937 | 0.4042 | 6.8 | 23.1 |
| MIMOUNet++ | 24.70 | 0.8894 | 0.4074 | 16.1 | 47.4 |
| NAFNet | 24.80 | 0.9047 | 0.4109 | 67.9 | 52.9 |
| Restormer | 24.90 | 0.9045 | 0.3875 | 26.1 | 116.9 |
| IFIRNN | 24.75 | 0.8984 | 0.3434 | 4.1 | 7.6 |
| DBN | 25.05 | 0.9050 | 0.3763 | 15.3 | 4.2 |
| EDVR | 25.15 | 0.9088 | 0.3773 | 23.6 | 320.1 |
| BasicVSR | 25.10 | 0.9117 | 0.2885 | 6.2 | 55.8 |
| BasicVSR++ | 25.60 | 0.9121 | 0.2207 | 9.5 | 57.1 |
| DSTNet | 25.70 | 0.9134 | 0.2604 | 7.5 | 22.1 |
| RefVSR | 25.58 | 0.9124 | 0.2156 | 4.8 | 268.3 |
| Ours | **26.34** | **0.9167** | **0.1614** | 8.8 | 248.7 |

convolutional layer to generate the final output directly. We meticulously re-train all baseline methods on our dataset to ensure a fair and comprehensive comparison using the publicly available code. However, due to *limited computational resources* (with only a 3090 GPU and 1080 Ti GPUs available), we cannot reproduce specific potential video deblurring methods. We plan to incorporate more advanced and computationally demanding methods when sufficient computational resources are available.

**Quantitative results.** As shown in Table 1, our proposed AsLeD-Net consistently achieves the best performance across PSNR, SSIM, and LPIPS metrics. Our method outperforms the second-best DSTNet by 0.64 dB in PSNR (26.34 vs. 25.70 dB), demonstrating superior noise reduction and detail preservation. Compared to widely-used methods like BasicVSR++ (25.60 dB) and EDVR

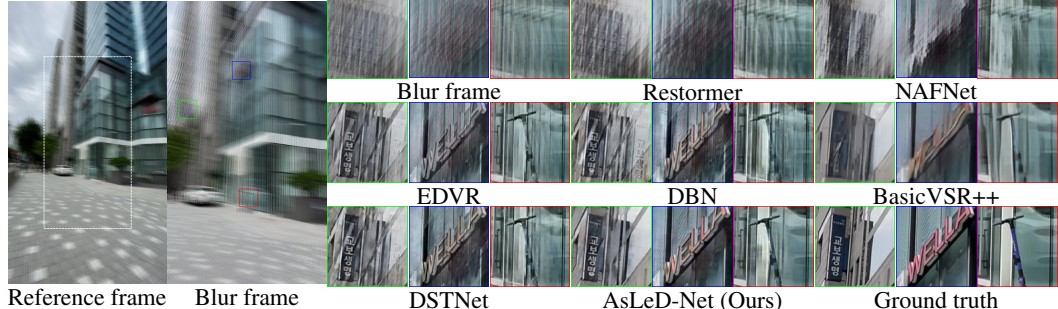

Figure 6: Qualitative comparison of AsLeD performance on the RealMCVSR dataset.

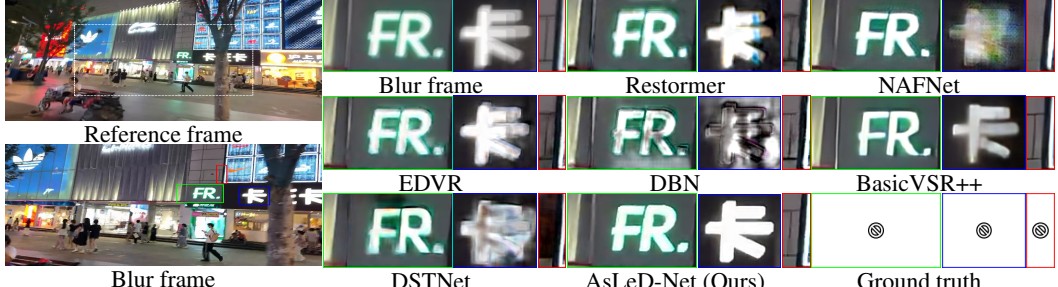

Figure 7: Qualitative comparison of AsLeD performance on real-world blurry scenes captured using an iPhone 14 Plus. *More results can be found in the supplementary material.*

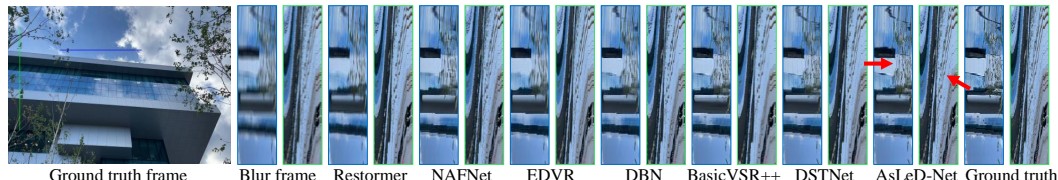

Figure 8: Temporal consistency comparison on the AsLeD task, where temporal profiles are extracted along horizontal (in blue) and vertical (in green) directions of the reconstructed frames.

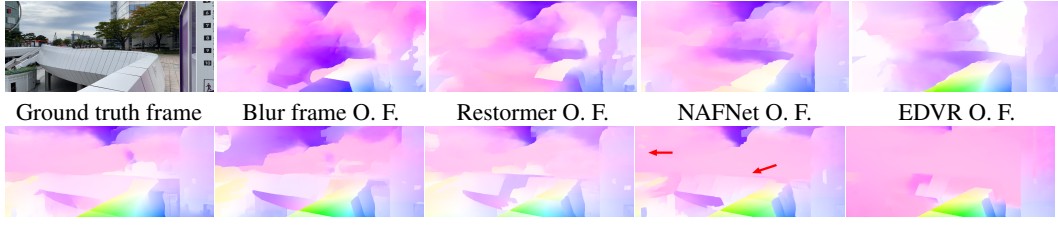

Figure 9: Temporal consistency comparison on the AsLeD task. The optical flow (O. F.) are estimated using the pre-trained RAFT [68].

(25.15 dB), our approach achieves gains of 0.74 dB and 1.19 dB, respectively. For SSIM, AsLeD-Net achieves 0.9167, improving upon DSTNet (0.9134), BasicVSR++ (0.9121), and RefVSR (0.9124), indicating enhanced structural consistency. Moreover, our method sets a new benchmark in LPIPS (0.1614), significantly surpassing the closest competitor, RefVSR (0.2156), with a 0.0542 reduction, demonstrating the perceptual quality of our restored frames.

**Computational cost results.** As shown in Table 1, our AsLeD-Net achieves superior performance on the RealMCVSR dataset but comes with a higher model complexity (8.8M parameters) and longer inference time (248.7 ms) compared to lightweight models like IFIRNN (4.1M, 7.6 ms) and DBN (15.3M, 4.2 ms). This limitation hinders its applicability in real-time or resource-constrained

scenarios. This limitation indicates that our method might not be optimal for real-time or resource-constrained applications. In future work, we plan to explore model compression techniques, such as knowledge distillation and lightweight architecture designs, to reduce computational costs while maintaining high restoration quality.

**Qualitative results.** We present visual comparison results on the RealMCVSR dataset in Figures 6 and 7. AsLeD-Net excels in preserving fine texture details, consistently outperforming methods such as Restormer, NAFNet, EDVR, DBN, DSTNet, and BasicVSR++. It effectively retains architectural details, such as grid-like window patterns and building facades, which are often blurred by other methods. As shown in Figure 7, AsLeD-Net produces sharper, more legible Chinese characters on signs and enhances clarity in real-world iPhone 14 Plus captures, revealing finer details in street signs and logos, thus improving text readability and recognition for practical applications.

**Temporal consistency.** We present the temporal consistency analysis in Figure 8 and Figure 9, highlighting the superior performance of our method in preserving motion coherence and fine details across consecutive frames. Compared to baselines, our method maintains smoother transitions and finer details in the temporal profiles, ensuring better motion consistency and reducing artifacts such as flickering. The estimated optical flow using RAFT [68] closely matches the ground truth and exhibits sharper, more well-defined structures than baselines. This demonstrates our model's ability to capture temporal coherence.

### 4.3 Ablation Study

We conduct experiments on RealMCVSR in terms of PSNR/SSIM. *Due to space limitations, please refer to the supplementary material.*

Table 2 ablates the three core components. Each single module improves the baseline (25.40 PSNR / 0.9120 SSIM). RMC yields the largest single module gain (25.79 / 0.9143). Adding RMC to ALM or DC brings further gains, and using all three achieves the best result (26.34 / 0.9167). This shows that temporal alignment is pivotal, while ALM and DC provide complementary spatial refinement. Within context aggregation, our ALM with $k = 3$ attains the best PSNR/SSIM

Table 2: Ablation study of three core components in AsLeD-Net on RealMCVSR. We replace the removed components with the residual blocks, ensuring parameter consistency.

| Method | Core Components | | | RealMCVSR | |
|---|---|---|---|---|---|
| | ALM | DC | RMC | PSNR↑ | SSIM↑ |
| (a) | ✗ | ✗ | ✗ | 25.40 | 0.9120 |
| (b) | ✓ | ✗ | ✗ | 25.72 | 0.9137 |
| (c) | ✗ | ✓ | ✗ | 25.69 | 0.9138 |
| (d) | ✗ | ✗ | ✓ | 25.79 | 0.9143 |
| (e) | ✗ | ✓ | ✓ | 25.89 | 0.9145 |
| (f) | ✓ | ✗ | ✓ | 25.84 | 0.9140 |
| (g) | ✓ | ✓ | ✗ | 25.75 | 0.9139 |
| (h) | ✓ | ✓ | ✓ | 26.34 | 0.9167 |

and outperforms DAT and MASA SR under the same protocol. For latency sensitive settings, an ALM+RMC variant offers a favorable speed and quality tradeoff. Notably, ALM+DC without RMC (25.75 / 0.9139) yields only limited uplift, indicating that temporal alignment is the dominant bottleneck under realistic motion. Taken together with the ALM study (where $k=3$ peaks), these results suggest that structure aware reference matching and temporal propagation mitigate complementary error modes, namely cross view misalignment and blur induced motion ambiguity, and that a reduced ALM+RMC configuration is a practical choice when latency is critical.

## 5 Conclusion

In this paper, we propose AsLeD-Net, a practical video deblurring method that leverages the complementary perspectives of asymmetric dual-lens systems. By aligning and propagating temporal reference features from ultra-wide views and fusing them with blurry wide-angle frames, AsLeD-Net effectively addresses cross-lens redundancy. Key modules such as the ALM module and the DC module ensure refined feature alignment and spatial consistency, while the RMC module enhances temporal alignment. Through extensive experiments, we validate the effectiveness of AsLeD-Net, showcasing its superiority over existing methods for deblurring asymmetric lens systems.

Our method performs well across various blur levels and scenarios, but frame averaging for training has limitations in simulating real-world motion blur, especially in high-motion and high-contrast scenes, and is frame rate-dependent; *see the supplementary material for details*.

**Acknowledgment.** This project is supported by the National Research Foundation, Singapore, under its Medium Sized Center for Advanced Robotics Technology Innovation.

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
