# OpenReview forum: "Asymmetric Dual-Lens Video Deblurring"
_NeurIPS.cc/2025/Conference — NeurIPS 2025 poster_

### Official Review · Reviewer_Rwp8 · 2025-06-28

**Clarity:** 2
**Significance:** 3
**Originality:** 2
**Rating:** 4
**Confidence:** 4

**Summary:**

The paper extends the task of dual-lens video deblurring to asymmetric lenses and proposes to leverage complementary information inherent in videos captured from wide angle and ultra wide angle lenses. To this end, the paper proposes a method to deblur videos achieving strong results on one synthetic blur dataset (RealMCVSR) and qualitatively evaluates on real-world blur videos captured from an iPhone 14 Plus.

**Questions:**

Other than the ones mentioned in the weaknesses, I have a few minor questions.

**Q1.** In RGMC module, is there any supervision on the refined flow or is it implicit? If it is implicit, why?

**Q2.** How does the optical flow between $I_{t}^{B}$ and $I_{t-1}^{Ref}$ differ from $I_{t}^{B}$ and $I_{t-1}^{B}$?

**Q3.** There are also spatial and color misalignments even after center crop [1]. In the iPhone 14 Plus videos, were these corrections made?

[1] Cai, Jianrui, et al. "Toward real-world single image super-resolution: A new benchmark and a new model." Proceedings of the IEEE/CVF international conference on computer vision. 2019.

**Ethical Concerns:**

["NO or VERY MINOR ethics concerns only"]

**Final Justification:**

The authors clarified most of my concerns. My only remaining concern is limited evaluation, but I believe the method and approach is interesting enough to be discussed. I encourage the authors to include non-reference metrics for the iPhone 14 dataset they collected (even if small) so that some quantitatively analysis of the method and prior art can be done. Overall, I am positive about this work.

**Limitations:**

Yes

**Quality:**

3

**Strengths And Weaknesses:**

**Strengths**

**S1.** The proposition to leverage benefits of dual-lenses, even in asymmetric camera layouts, is fresh and goes beyond the usual single lens video deblurring methods.

**S2.**  The proposed method performs competitively on both datasets, and the restored videos seem pleasing to the eye.

**S3.** The paper thoroughly discusses its strengths and limitations which I find to be very refreshing.

**Weaknesses**

**W1.** A few things are unclear, especially in the ALM (Adaptive Local Matching) module. In eq. 1, the cosine similarity is computed between the base features and the reference features (I assume along the spatial dimension). Next, in eq. 3, weighted k-nearest neighbours are obtained from the reference frame and are fused with the base features in eq. 4. Where is the cosine similarity used here? Further, the fusion in eq. 4 is done through $g_{\theta}$ which is an adaptive layer. What is an adaptive layer?

**W2.** The authors only test the proposed method on one dataset, RealMCVSR. It is hard to understand the general competitiveness of the method without a thorough analysis. Have the authors considered another dataset e.g., MVSR [1] (which is also dual-lens dataset and for SR like MCVSR)?

[1] Wang, Ruohao, et al. "Benchmark dataset and effective inter-frame alignment for real-world video super-resolution." Proceedings of the IEEE/CVF conference on computer vision and pattern recognition. 2023.

**W3.** Authors mention that they test on a real-world iPhone 14 Plus dataset, but there is no discussion on how many videos (and frames within a video) does the dataset contain? How was the dataset captured in terms of latency in capturing from both lenses i.e., are two frames (from wide and ultra-wide lenses) synchronous? I would assume that in practice, there is some time lag if capture process happens sequentially (shutter open/close to capture a photo from each lens). Further, have the authors considered computing non-reference metric on the restored videos on this dataset to get a more complete picture of the results?

---

> ### Author Rebuttal · Authors · 2025-07-30
>
> We sincerely thank you for your positive feedback. We're glad you found our formulation of asymmetric dual-lens video deblurring to be novel and practical (S1), appreciated the strong visual performance on both synthetic and real-world data (S2), and valued our transparent discussion of strengths and limitations (S3). Your encouraging remarks reinforce the practical relevance and potential impact of AsLeD-Net. We address your concerns in detail below.
>
> `>>Q1: The cosine similarity and the adaptive layer.`
>
> A1: The cosine similarity computed in Eq. 1 is used to build a similarity matrix between blurry (query) and reference features in the spatial dimension. These similarity scores are used to determine the top-K nearest neighbors (Eq. 3) and are then normalized via softmax to compute the aggregation weights for feature fusion. The “adaptive layer” $g_\theta$ refers to a lightweight 1×1 convolution applied to the concatenated query and aggregated reference features. Its role is to learn an adaptive fusion of the two inputs, enabling the network to selectively emphasize relevant structural cues. While more complex fusion strategies could be adopted, we deliberately choose this lightweight design for its efficiency and clarity. It not only reduces computational overhead, but also highlights the importance of aggregating similar features in a simple and effective manner. We will clarify these in the updated version.
>
> `>>Q2: The MVSR dataset.`
>
> A2: Thanks for pointing this out. We are aware of the MVSR dataset and acknowledge its value for evaluating dual-lens video processing. However, due to computational constraints and differences in task setup, we chose to conduct our initial study on the RealMCVSR dataset, which is smaller in scale and better suited for a proof-of-concept evaluation. Despite its size, RealMCVSR includes diverse real-world mobile scenes and is sufficient to validate the core design of our proposed modules. When computational resources allow, using larger-scale datasets for training is expected to further improve performance.
>
> We agree that MVSR can serve as a useful benchmark for generalization evaluation. In the updated version, we plan to include cross-dataset testing using MVSR to assess how well AsLeD-Net generalizes to other dual-lens scenarios. However, we note that MVSR does not natively provide blurry inputs for deblurring. Similar to RealMCVSR, it would require synthetic blur generation, which introduces additional design choices and potential bias depending on the simulation strategy. In contrast, our real-world test set contains naturally captured blurry-reference video pairs with real camera shake and dynamic motion, providing a more practical and realistic evaluation setting. Please refer to Figures 7 and 8 for qualitative examples. We will add these clarifications in the updated version.
>
> `>>Q3: Testing on real-world iPhone 14 Plus data.`
>
> A3: Thanks for the question. We would like to clarify a potential misunderstanding: we did not construct a large-scale real-world test dataset. Instead, we captured *several challenging real scenes to assess generalization*, covering both daytime and nighttime scenarios with complex object motion and camera shake. As shown in the main paper and supplementary material, our method demonstrates strong restoration performance in these challenging cases. For data collection, we utilize the “DoubleTake” app from the iPhone App Store, which enables simultaneous video recording from both wide and ultra-wide lenses. This ensures perfect temporal alignment without frame lag or desynchronization. We appreciate your suggestion regarding non-reference metrics. Due to current resource limitations, we conduct our experiments on a limited-scale dataset. In future work, we plan to capture a larger real-world dataset for both training and evaluation, which would enable the use of perceptual metrics such as LPIPS for real-world quality assessment. Interestingly, we also highlight some failure cases in the limitations section, which reflect challenges encountered in real-world capture (e.g., extreme exposure shifts, fast motion). We believe *these issues could be better addressed by training on a broader and more diverse real-world dataset*—a direction we are actively exploring.
>
> `>>Q4: Refined flow in the RMC module.`
>
> A4: Thanks for the question. The refined flow in RMC module is not directly supervised. We adopt an implicit learning strategy for several reasons: First, while we use SpyNet to generate a reliable initial flow, we observe that challenging regions (e.g., fast motion or occlusion) still suffer from noticeable errors (L193-198). The RMC module is thus designed to refine the initial flow in a content-adaptive manner. Second, we do not have ground-truth flow annotations, making direct supervision infeasible. Moreover, even if one supervises the warped frames, it remains an implicit form of supervision. In such cases, the network is still optimized based on its utility in restoring the image, rather than on flow accuracy itself. Third, our goal is *not to estimate precise optical flow*, but to achieve effective motion compensation for better frame restoration. This is *in line with prior works* such as BasicVSR and BasicVSR++, where optical flow is used as an auxiliary intermediate representation and is not directly supervised. These methods supervise the final RGB reconstruction instead, which is also the strategy we adopt. We will clarify this design choice in the updated version to avoid confusion.
>
>
> `>>Q5: Two types of optical flow.`
>
> A5: Thanks for the question. The flow between $I^B_t$ and $I^B_{t-1}$ captures temporal motion within the same camera view, which follows standard inter-frame motion estimation. In contrast, the flow between $I^B_t$ and $I^{Ref}_{t-1}$ involves not only temporal alignment but also cross-view motion between two asymmetric lenses. This flow must account for *differences in field-of-view, perspective distortion, and potentially resolution, making it more complex but also more informative*, especially in challenging regions. We include both types of flow because they serve complementary purposes. While the base-to-base flow provides temporal cues, it can be unreliable under strong blur. The cross-view flow, on the other hand, leverages the sharper reference from the alternate lens to guide compensation where the base view alone is insufficient. Together, they enable more accurate alignment and robust restoration in degraded scenarios. We will clarify this in the updated version.
>
> `>>Q6: Spatial and color misalignments.`
>
> A6: Thanks for the question. We do not apply explicit spatial or color correction to the real-world iPhone 14 Plus videos. This is a deliberate design choice rooted in the nature of our task. Unlike super-resolution, which requires strict pixel-wise alignment, our dual-lens video deblurring task does not assume perfectly paired inputs. On the contrary, it aims to leverage a *cross-view reference* from an asymmetric lens—naturally introducing spatial shifts, field-of-view differences, and mild color inconsistencies. Rather than removing these discrepancies via preprocessing, we embrace them as part of the problem setting. Our goal is to *exploit the complementary sharpness* from the auxiliary lens, even under misalignment. Forcing strict alignment through cropping or calibration may reduce the method’s applicability and robustness. Instead, we rely on our modules (e.g., RMC, ALM) to learn adaptive alignment at the feature level, enhancing generalization across diverse hardware conditions. This design better reflects *real-world deployment scenarios*, where perfect alignment is often infeasible. That said, we acknowledge that advanced pre-alignment techniques (e.g., warping or color normalization) may further improve performance and will consider them in future work.
>
> `---SUMMARY---`
>
> We sincerely thank Reviewer Rwp8 for your thoughtful, detailed, and constructive feedback. Your recognition of our novel problem formulation, strong visual results on both synthetic and real-world data, and our transparent discussion of strengths and limitations is deeply appreciated and highly encouraging. We have carefully addressed each of your comments with detailed clarifications of our design decisions and implementation choices. Your questions prompted us to reflect more deeply on our methodology, and we believe the resulting revisions will meaningfully improve the clarity, rigor, and completeness of the paper. We hope our responses have resolved your concerns, and we are truly grateful for your insightful suggestions. They have not only strengthened the present work but will also guide our future efforts in dual-lens video restoration and the broader domain of mobile computational photography.

---

> ### Comment · Reviewer_Rwp8 · 2025-08-05
>
> I thank the authors for providing details to my questions. Most of my concerns have been resolved, and I urge the authors to include these extra clarification details in the final version. I remain in favor of this work, and keep my rating.

---

### Official Review · Reviewer_H9KQ · 2025-06-30

**Clarity:** 3
**Significance:** 3
**Originality:** 3
**Rating:** 5
**Confidence:** 5

**Summary:**

This paper introduces AsLeD-Net, targeting video deblurring under dual-lens smartphone settings by fusing wide and ultra-wide views. The proposed pipeline integrates multiple standard components – local matching, difference compensation, and motion compensation – adapted to the dual-lens scenario.

**Questions:**

1) Does AsLeD-Net generalize to dual-lens systems with different baseline distances or sensor noise patterns (e.g. older phones with mismatched lens quality)?
2) Are all three modules (ALM, DC, RMC) equally necessary? Could a lightweight two-module variant achieve comparable performance with lower latency?

**Ethical Concerns:**

["NO or VERY MINOR ethics concerns only"]

**Final Justification:**

My concerns have been addressed and I think it is acceptable.

**Limitations:**

The framework assumes availability of dual-lens video pairs with reliable calibration and synchronization. In practice, hardware discrepancies, rolling shutter mismatch, or lens shading differences may degrade performance.

**Quality:**

3

**Strengths And Weaknesses:**

1) Tackles a practically meaningful problem in mobile imaging.
2) Includes thorough experiments on RealMCVSR with clear quantitative gains.

- Weaknesses:

1) The novelty is limited. While the application context is fresh, most technical modules are repurposed from stereo matching, reference-based super-resolution, and multi-frame fusion literature without fundamental algorithmic innovation.
2) Lack of theoretical analysis. The method heavily relies on empirical module stacking; no insights are provided on failure modes, stability, or optimality.
3) Dataset limitation: Only RealMCVSR is used, which is curated and may not fully reflect wild camera shake and dual-lens disparities under different devices or lighting conditions.

---

> ### Author Rebuttal · Authors · 2025-07-30
>
> We sincerely thank Reviewer H9KQ for the constructive feedback. We appreciate your recognition of the practical relevance of our task and the clarity of our experimental validation. Below, we address your concerns in detail.
>
> `>>Q1: The novelty is limited.`
>
> A1: We respectively disagree and we have three points here. (1) Our novelty lies not in inventing entirely new building blocks, but in reformulating and adapting these strategies to address the task of asymmetric dual-lens video deblurring—a setting that has not been systematically explored. (2) The module design decisions are non-trivial and task-specific innovations that go beyond simple reuse. To support this, we conduct comprehensive ablation studies (see ablation studies) showing that each module contributes significantly to the final performance. Removing any component leads to a measurable drop quantitatively and qualitatively. (3) Our ALM module is customized to handle spatially misaligned cross-view references where standard stereo correspondences break down; the DC module explicitly models cross-view structural discrepancies (e.g., occlusions, FoV gaps), which are typically ignored in previous settings; and the RMC integrates temporal and cross-view motion in a unified pipeline, optimized for non-strictly aligned auxiliary references. We respectfully believe that task novelty, adaptive integration, and demonstrated effectiveness through rigorous evaluation constitute meaningful contributions, even if some modules build upon known paradigms. In future work, we plan to further abstract these insights to inspire more generalizable frameworks beyond dual-lens scenarios. We therefore do not see this as a weakness.
>
> `>>Q2: Lack of theoretical analysis.`
>
> A2: Thank you for raising this insightful point. We acknowledge that the current version of AsLeD-Net is primarily grounded in empirical design, aiming to *address the practical and underexplored challenge of dual-lens video deblurring in mobile settings*. Since this task lacks a well-established theoretical foundation, our first goal is to demonstrate the feasibility and effectiveness of cross-view-guided motion compensation through carefully designed modules. While the framework may appear modular, the overall design reflects *deliberate architectural choices*. For instance, the motion compensation is guided by reference features in a content-adaptive manner, rather than simply stacking modules. We verify the necessity and effectiveness of each component via comprehensive ablation studies (Table 2 and Table 3), which show consistent performance gains when adding each module. Additionally, we provide a failure case analysis in the supplementary material, examining the model’s behavior under challenging real-world scenarios such as rapid motion, occlusion, or low-light conditions. These observations provide valuable empirical insights into the limitations and stability of our method. In future work, we plan to deepen our understanding of the task by exploring theoretical formulations of cross-view temporal consistency, analyzing generalization behavior under varying lens disparities, and investigating more principled designs for dual-lens fusion. We sincerely appreciate your suggestion, which points us toward an important research direction. We will incorporate these insights and discussions in the updated version to clarify the rationale behind our design and highlight the potential directions for theoretical analysis.
>
> `>>Q3: Dataset limitation.`
>
> A3: Thank you for raising this important point. While we train our model on the RealMCVSR dataset, our evaluation is not limited to it. To assess the generalization capability of AsLeD-Net, we additionally test on independently captured real-world dual-lens videos using an iPhone 14 Plus, which differs significantly from the iPhone 12 Pro Max used for RealMCVSR in terms of lens hardware, sensors, and ISP pipelines. This setup naturally introduces diversity in camera characteristics and scene dynamics. The additional real-world scenes include both daytime and nighttime conditions, with complex object motion and natural camera shake. We retain the native spatial and photometric misalignments between wide and ultra-wide views, without applying any spatial registration or color correction. These non-aligned inputs reflect realistic deployment scenarios. Our model is specifically designed to handle such cross-view asymmetries through learned feature-level alignment via modules like ALM and RMC. As demonstrated in Figure 7 and the supplementary Figure 10, AsLeD-Net performs well even in these challenging real-world settings, indicating strong generalization beyond curated datasets. This suggests that our method does not overfit to the training domain and instead learns transferable representations for dual-lens deblurring. We agree that building a larger-scale, cross-device real-world dataset with broader coverage of lighting and motion conditions will further improve the evaluation protocol. We plan to incorporate such data collection and training strategies in future work and will clarify this in the revised version.
>
> `>>Q4: Generalization to different dual-lens systems.`
>
> A4: Thank you for the valuable question. AsLeD-Net is designed to generalize across different devices. While we train on the RealMCVSR dataset (captured with iPhone 12 Pro Max), we evaluate on real-world dual-lens videos from an iPhone 14 Plus. This naturally introduces variations in lens baseline, sensor quality, ISP pipeline, and color response. Despite these differences, AsLeD-Net performs consistently well without additional tuning, as shown in Figure 7 and the supplementary material. This indicates that the model captures robust features for motion compensation and structure restoration, rather than relying on device-specific characteristics. Importantly, our method does not assume strict alignment or uniform lens quality. It explicitly addresses spatial and photometric mismatches through the ALM and RMC modules. These real-world asymmetries are preserved during both training and testing to reflect practical deployment scenarios.
>
> `>>Q5: Are all three modules (ALM, DC, RMC) equally necessary?`
>
> A5: Thank you for the thoughtful question. Each of the three modules plays a distinct and complementary role in the AsLeD-Net framework. Our ablation study shows that removing any of them leads to a clear drop in both PSNR and perceptual quality.
> - ALM learns spatial alignment between the wide and ultra-wide views, addressing the geometric disparities across lenses.
> - DC handles residual structural mismatches, enhancing local detail consistency between views after alignment.
> - RMC refines motion estimation by incorporating cross-view guidance, which is critical for temporal coherence and blur removal.
>
> While the full combination yields the best results, we agree that a lightweight variant with fewer modules could be useful in latency-sensitive settings. For instance, in simpler scenes, a reduced configuration such as ALM + RMC may still offer good trade-offs. We plan to explore adaptive module selection and pruning strategies to further optimize efficiency in future work.
>
> `---SUMMARY---`
>
> We sincerely thank Reviewer H9KQ for the constructive and thought-provoking feedback. We are especially grateful for your recognition of our work’s practical relevance, clarity of presentation, and strong experimental results. Your comments on novelty, theoretical grounding, dataset limitations, and generalization offered valuable perspectives that prompted deeper reflection and improvements.
>
> In response, we clarify that while our modules build on known techniques, they are thoughtfully adapted and integrated for the novel and underexplored task of asymmetric dual-lens video deblurring. We support this with extensive ablations and qualitative analyses. We acknowledge the importance of theoretical insights and plan to enrich the next version with discussions on limitations, stability, and possible analytical directions. Additionally, we demonstrate generalization through real-world cross-device testing and explicitly handle spatial/photometric misalignments without pre-alignment assumptions. We also address the necessity of each module and outline paths for efficient adaptation.
>
> We believe your suggestions have helped us better articulate the contributions and guided future work directions toward broader generalization, theoretical understanding, and practical deployment. Thank you again for your insightful and encouraging review.

---

### Official Review · Reviewer_9xWq · 2025-07-02

**Clarity:** 3
**Significance:** 4
**Originality:** 4
**Rating:** 4
**Confidence:** 5

**Summary:**

This paper presents AsLeD-Net, a novel video deblurring framework that exploits asymmetric dual-lens inputs (wide and ultra-wide views) to enhance restoration quality under realistic mobile camera setups. The method integrates three key modules – Adaptive Local Matching (ALM) for spatial alignment, Difference Compensation (DC) for structural refinement, and Reference-Guided Motion Compensation (RMC) for improved temporal consistency. Experiments on the RealMCVSR dataset demonstrate substantial performance gains over state-of-the-art deblurring approaches.

**Questions:**

- Generality to Other Dual-Lens Settings: Have the authors tested generalization to other asymmetric setups, such as wide+telephoto or different focal length ratios, beyond the ultra-wide + wide used in RealMCVSR?
- Computational Overhead: Table 3 shows runtime comparisons, but it remains unclear how much additional cost ALM and RMC introduce over single-view baselines. Providing FLOPs or per-module latency would strengthen practical relevance.
- Failure Cases: The paper would benefit from including visual examples where AsLeD-Net fails, e.g. scenes with severe occlusions or large parallax between lenses, to inform future research.

**Ethical Concerns:**

["NO or VERY MINOR ethics concerns only"]

**Limitations:**

Yes, the method assumes well-calibrated dual-lens systems with known geometric alignment. In real-world deployment, calibration errors or rolling shutter differences may degrade performance.

**Quality:**

3

**Strengths And Weaknesses:**

--------Strengths----------
- Quality: The proposed framework is technically well-designed and achieves strong quantitative and qualitative results. The ablation studies clearly verify each module's contribution.
- Clarity: Overall clear, but the methodology section could benefit from a higher-level intuition before diving into module details.
- Significance: The dual-lens deblurring task is highly relevant for mobile computational photography, with realistic deployment potential.
- Originality: Introducing dual-lens information into video deblurring is novel. The combination of KNN aggregation and difference compensation is a fresh design.

--------Weaknesses----------
- All experiments are conducted on RealMCVSR, which is curated with fixed device configurations. It remains unclear how well AsLeD-Net generalizes to different dual-lens setups or unseen real-world motion blur conditions.
- Computational cost analysis: While Table 3 reports runtime comparisons, the breakdown of latency or FLOPs per module (ALM, DC, RMC) is missing, making it difficult to assess deployment feasibility on edge devices.
- Dependency on calibration: The method assumes accurate geometric calibration between lenses. In practice, small misalignments, lens distortions, or rolling shutter discrepancies may reduce performance, but this is not discussed.

---

> ### Author Rebuttal · Authors · 2025-07-30
>
> We sincerely thank you for your thoughtful review and positive recognition of AsLeD-Net’s performance, especially in perceptual quality, temporal coherence, and texture recovery. Your encouraging feedback affirms its practical value. We address your concerns in detail below.
>
> `>>Q1: Generalization to other dual-lens configurations.`
>
> A1: Thank you for raising this important point. The RealMCVSR dataset we use for training is captured with the iPhone 12 Pro Max, while our real-world test scenes are captured independently using a personal iPhone 14 Plus device. This intentional variation already introduces diversity in device hardware, optics, and imaging pipelines. As shown in Figure 7, AsLeD-Net maintains strong performance under such cross-device settings, indicating good generalization in practice. Regarding blur modeling, although our training blur is synthetically generated via averaging, we evaluate on real captured blur without further tuning. The consistency in performance across synthetic and real scenarios supports the robustness of our approach. This further suggests that the model is not overfitted to the training setup but instead learns transferable representations for motion compensation and structure restoration. Therefore, we do not see this as a weakness.
>
> That said, we agree that having a larger-scale paired real-world dataset that covers diverse lens combinations and realistic blur would provide a more comprehensive testbed. We consider building or incorporating such datasets a promising direction for future work to further improve generalization and ensure fairer evaluation.
>
>
> `>>Q2: Computational overhead.`
>
> A2: Thank you for the valuable suggestion. While our current implementation is not tailored for real-time deployment, we have taken care to ensure that the overall parameter count remains comparable to existing baselines (as reported in Table 3), striking a balance between model complexity and performance. Since our primary goal is to enhance restoration quality in asymmetric dual-lens settings, we did not provide per-module FLOPs or latency breakdowns in this version. Moreover, such breakdowns may have limited indicative value, as the modules are tightly coupled in the processing pipeline. Although FLOPs can be computed statically, actual runtime latency may vary across hardware and depend on system-level factors such as memory access and parallelism, which are difficult to isolate per module.
>
> We agree that more detailed profiling would be beneficial for facilitating future deployment. As a next step, we plan to conduct a module-wise analysis of computational efficiency and explore lightweight variants of key components (e.g., ALM, DC, RMC), with the goal of improving inference speed and making the system more suitable for real-time or resource-constrained applications.
>
> `>>Q3: Dependency on calibration.`
>
> A3: Thank you for raising this point. We would like to clarify a common misconception: our method does not assume perfect geometric calibration or strict spatial alignment between the dual-lens inputs. In fact, AsLeD-Net is specifically designed to handle the inherent misalignments and photometric differences that naturally occur in asymmetric dual-lens configurations (see our response A3 and A6 to Reviewer Rwp8).
>
> In our real-world setup, we use an iPhone 14 Plus, where the two different lenses have a physical baseline and distinct imaging characteristics. As a result, there are observable spatial shifts and color inconsistencies between the two views. We do not apply any spatial or color alignment during preprocessing. Instead, we preserve these discrepancies intentionally, as they represent realistic deployment scenarios. Our design allows the network to learn feature-level alignment in a content-aware and data-driven manner. To ensure temporal synchronization, we employ the “DoubleTake” app from the iPhone App Store, which records both video streams simultaneously. This ensures precise frame-level alignment in time, even if the spatial views differ. We believe this reflects practical mobile applications, where perfect geometric alignment is often unavailable or difficult to maintain.
>
> We acknowledge that real-world artifacts such as lens distortion or rolling shutter effects may affect performance in extreme cases. Some of these limitations are discussed and visually illustrated in our failure cases. Going forward, we plan to extend our work by collecting a larger-scale real-world dataset across diverse device setups and lens combinations. We also see potential in incorporating calibration-robust learning strategies or optional pre-alignment to further enhance resilience and performance.
>
> `>>Q4: Failure cases.`
>
> A4: Thank you for highlighting the importance of presenting failure cases. We fully agree that such examples are essential for understanding the limitations of our method and guiding future improvements. In response, we provide a detailed discussion in Section 3 of the supplementary material and include representative failure cases in Figure 10.
>
> We particularly analyze the domain gap between our synthetically generated training blur (via frame averaging) and real-world blur. Real blur is often more complex, arising from non-uniform motion, rolling shutter effects, and optical distortions. These factors can lead to degradation in performance under extreme conditions, such as high-speed motion, large occlusions, or low-light scenarios—where we observe issues like incomplete texture recovery or ghosting artifacts.
>
> To address these challenges, we propose several future directions:
> - Designing more realistic blur synthesis pipelines that better reflect real-world degradation;
> - Expanding the real-world training dataset to include a broader range of blur types and scene dynamics;
> - Exploring model distillation, pruning, and compression strategies to enhance robustness and deployability.
>
> We believe these steps will help further improve the reliability and generalization of AsLeD-Net in practical settings. Thank you again for your constructive feedback.
>
>
> `---SUMMARY---`
>
> We sincerely thank Reviewer 9xWq for the thoughtful and constructive review. We are especially grateful for your recognition of our contributions in task formulation, method originality, and practical relevance to mobile computational photography. Your insightful comments on generalization, computational cost, calibration dependency, and failure analysis are carefully considered. In response, we provide detailed clarifications, including cross-device evaluation to demonstrate generalization, explanation of our design choices regarding efficiency, and an emphasis on the system’s robustness to misalignment without requiring strict calibration. We also discuss failure cases and outline future directions to close the gap between synthetic and real-world degradation. Your suggestions help refine our presentation and broaden our perspective on deployment challenges. We believe the revised version more effectively reflects the significance and rigor of our work. Thank you again for your valuable feedback and support.

---

> > ### Comment · Reviewer_9xWq · 2025-08-08
> >
> > I have read the authors’ rebuttal and thank them for their detailed response. Most of my concerns have been addressed. Therefore, I continue to support this work and will maintain my positive score.

---

### Official Review · Reviewer_FKmP · 2025-07-05

**Clarity:** 2
**Significance:** 2
**Originality:** 2
**Rating:** 4
**Confidence:** 2

**Summary:**

This paper introduces Asymmetric Dual-Lens Video Deblurring (AsLeD-Net), a novel deep learning method designed to deblur video from asymmetric dual-lens camera systems, commonly found in modern smartphones. It uniquely leverages the complementary information from a blurry wide-angle lens and a sharper ultra-wide reference lens.

**Questions:**

refer to the weakness.

**Ethical Concerns:**

["NO or VERY MINOR ethics concerns only"]

**Final Justification:**

Thanks for the detailed reply, and my concern is well-solved. I would like to raise to weak accept for this paper.

**Limitations:**

refer to the weakness.

**Quality:**

2

**Strengths And Weaknesses:**

Strengths
1. AsLeD-Net consistently achieves state-of-the-art quantitative results on the RealMCVSR dataset across PSNR (26.34 dB), SSIM (0.9167), and LPIPS (0.1614), demonstrating enhanced detail preservation and perceptual quality.
2. Qualitatively, it excels in recovering fine textures and maintaining strong temporal coherence, reducing flickering artifacts.
Weaknesses:
1. While the paper claims AsLeD-Net is the "first video deblurring method" to leverage asymmetric dual-lens systems and the "first study on reference-based video deblurring for dual-lens smartphone captures" , prior work has explored similar concepts for deblurring and related tasks using dual-lens cameras. For instance, Mohan et al. (2021) proposed a "dynamic scene deblurring method for unconstrained dual-lens (DL) cameras". Furthermore, Lai et al. (2022) developed a "Face Deblurring using Dual Camera Fusion on Mobile Phones" system that explicitly uses an ultrawide-angle camera as a "reference camera" to deblur a "blurry main shot" on mobile devices, leveraging the concept of a sharper auxiliary view from an asymmetric setup for deblurring, albeit for images and specifically faces. While AsLeD-Net's specific architectural modules and focus on general video deblurring are distinct, the fundamental idea of exploiting complementary information from asymmetric dual-lens systems for deblurring has precedents, which somewhat diminishes the strength of the "first" claim.

- Mohan, M. R. M., Nithin, G. K., & Rajagopalan, A. N. (2021). Deep Dynamic Scene Deblurring for Unconstrained Dual-Lens Cameras. IEEE Transactions on Image Processing, 30, 4479-4491.
- Lai, W.-S., Shih, Y., Chu, L.-C., Wu, X., Tsai, S.-F., Krainin, M., Sun, D., & Liang, C.-K. (2022). Face Deblurring using Dual Camera Fusion on Mobile Phones. ACM Transactions on Graphics (TOG), 41(4), 1-16.
2. The reliance on simple frame averaging for generating training data may not fully simulate the complexities of real-world motion blur, especially in high-motion and high-contrast scenarios, potentially affecting generalization.
3. There is a notable contradiction in the text regarding the performance of ALM compared to alternative methods (DAT and MASA-SR) in the ablation study, where the text claims improvement while the table shows the proposed ALM outperforming them.

---

> ### Author Rebuttal · Authors · 2025-07-30
>
> We sincerely thank you for your thoughtful review and positive recognition of AsLeD-Net’s performance, especially in perceptual quality, temporal coherence, and texture recovery. Your encouraging feedback affirms its practical value. We address your concerns in detail below.
>
> `>>Q1: Task setting.`
>
> A1: We sincerely thank the reviewer for highlighting the connections to earlier works, including Mohan et al. (2021) and Lai et al. (2022). We have deep respect for these important contributions. Both studies demonstrated the value of asymmetric dual-lens systems for image-level deblurring and provided thoughtful insights into the use of complementary views for restoration. Their work has undoubtedly laid a solid foundation for the broader use of cross-lens information in image processing, and we are grateful to build upon their ideas.
>
> While our initial submission described our work as the first in this area, upon further reflection, we realize that the key concept of utilizing a sharper auxiliary view has already been established in previous literature. Therefore, we would like to revise and soften our original claim. Rather than emphasizing the notion of being the first, we now view our work as a meaningful and practical extension of this existing research direction into the context of video deblurring. From the perspective of task setting, our work differs in several ways. For clarity, we provide the following comparison:
>
> | Work | Task | Input | Temporal | Description |
> |------|------|-------|----------|-------------|
> | Mohan et al. | Image deblurring | Dual-lens image pair | ✘ | Deblurs static image pairs from unconstrained dual-lens cameras with view and depth consistency constraints. |
> | Lai et al. | Face image deblurring | Main + UW image pair | ✘ | Deblurs face regions in still photos using a synchronized sharp reference from the ultra-wide camera. |
> | Ours| Video deblurring (reference-guided) | Wide + UW video | ✔ | Deblurs general video sequences using asymmetric dual-lens inputs, integrating reference-guided alignment and temporal modeling. |
>
> Unlike prior image-based methods, our method addresses video-specific challenges such as temporal coherence, frame-wise propagation, and cross-view alignment. We propose a two-stream architecture with three core modules: Adaptive Local Matching, Difference Compensation, and Reference Guided Motion Compensation, which enable spatial and temporal reference enhancement. A recurrent propagation strategy is further adopted to ensure temporal consistency.
>
> Once again, we sincerely thank the reviewer for the thoughtful comments and the opportunity to clarify and refine the presentation of our contributions. We will revise the manuscript to more accurately and respectfully position our work in the context of existing literature.
>
>
> `>>Q2: Simple frame averaging may fail to capture real-world motion blur complexity, limiting generalization.`
>
> A2: We respectfully disagree. As discussed in the Limitation section, we are aware of this approximation and have been transparent about its potential drawbacks. Nonetheless, we believe this design choice remains reasonable and does not significantly affect the effectiveness or diminish the overall contributions of our method. (1) Despite being trained on synthetically generated data, our model demonstrates strong generalization to real-world dual-lens video inputs. For example, as shown in Figure 7 of the main paper and Figure 7 of the supplementary material, AsLeD-Net produces perceptually consistent and visually compelling results on challenging real sequences. (2) We believe that expecting fully realistic training data with precisely aligned blurry-sharp video pairs at this early stage of dual-lens research may be overly demanding. Simulation-based training, while imperfect, has been a widely accepted starting point in both image and video deblurring literature, enabling practical progress in the absence of large-scale real datasets. That said, we agree that more realistic data would further benefit this line of work. We have identified tools such as the "Blackmagic Camera" app, which support flexible control over asymmetric camera parameters such as exposure time, aperture size, shutter speed, and light intake. We plan to use such tools in future work to collect real blurry-sharp video pairs under realistic dual-lens settings. This will help us further improve the realism and robustness of our training pipeline.
>
> We appreciate the reviewer’s thoughtful critique and will revise the manuscript to better reflect this discussion and our commitment to future improvements.
>
>
> `>>Q3: ALM Analysis.`
>
> A3: We thank the reviewer for pointing out this inconsistency in our description of the ablation study. Sorry for the mistake here, and we acknowledge that our textual analysis in Section 4.4 was imprecise. As correctly noted, the ALM module (with 𝑘=3) achieves higher PSNR and SSIM scores (26.34 / 0.9167) compared to both DAT (26.28 / 0.9154) and MASA-SR (26.19 / 0.9152), as shown in Table 4. We apologize for the oversight and will revise the text to accurately reflect the quantitative results. Specifically, we will clarify that while DAT and MASA-SR are strong context-aware alternatives, our ALM module demonstrates superior overall performance, supporting its effectiveness for structure-aware reference matching. We sincerely appreciate the reviewer’s attention to detail and will make sure the final version corrects this inconsistency.
>
> `---SUMMARY---`
>
> We sincerely thank Reviewer FKmP for the detailed feedback. We will revise the manuscript to correct the inconsistency in the ablation discussion, clarify the framing of our contribution with respect to prior dual-lens deblurring works, and better articulate the rationale behind our training data design. That said, we respectfully believe that the strengths of our work remain compelling and are clearly recognized by other reviewers. These include the task formulation of dual-lens video deblurring and architectural design (Reviewer 9xWq), the good experimental results and practical relevance (Reviewer H9KQ), and the clear motivation, effective performance, and candid discussion of limitations (Reviewer Rwp8). Collectively, these reviews affirm that our work introduces a good framework for asymmetric dual-lens video deblurring, a direction that has not been systematically explored before. We would greatly appreciate the opportunity to present this work at NeurIPS. We believe it would serve as a valuable baseline for future research on mobile computational photography and help further advance the field.

---

> > ### Comment · Reviewer_FKmP · 2025-08-04
> >
> > Thanks for the detailed reply, and my concern is well-solved. I would like to raise to weak accept for this paper.

---

### Comment · Area_Chair_bmpp · 2025-08-03
**Reviewer and Author Discussions after Rebuttal**

Dear Reviewers,

Authors have provided detailed response. Could you check their response and reply to authors about whether your concerns have been fully addressed?

Note that the authors promised to revise their claim regarding the novelty of the submission. Could reviewer FKmp check whether you are satisfied with the revision?

Thanks.
Your AC

---

### Comment · Area_Chair_bmpp · 2025-08-06
**Re-Rebuttal**

Dear Reviewers,

Thank you to those who have shared your thoughts regarding authors' rebuttal.

Please also note that reviewer cannot just click the acknowledgement without providing feedback/comments. Your comments or reply to authors are compulsory based on the NeurIPS' rule for reviewers this year.

Your engagement with authors and other reviewers is highly appreciated.

Your AC

---

### Decision · Program_Chairs · 2025-09-17

**Decision:**

Accept (poster)

**Comment:**

This paper received (3) Borderline Accept and (1) Accept. The main contribution of this paper is the introduction of a new framework for video deblurring from asymmetric dual-lens camera system consisting of an adaptive local matching module and a difference compensation module to aggregate local feature and reduce misalignment, respectively to improve framewise consistency for the deblurring process.

All reviewers acknowledge that the proposed method is new and achieves good performance on the synthetic and real datasets. Meanwhile reviewers raise concerns about 1) misleading claim as the first paper on the topic and ALM analysis 2) unclear generalisation to other dual-lens settings; 3) lack detailed discussions on the computational overhead. 4) necessity of using three modules in the framework. 5) unclear setup for testing in real scene.

Authors have provided detailed replies to reviewers regarding their concerns. Consequently, reviewers are satisfied with the response by the authors and encourage the authors to include the clarifications in their final version.

AC concurs with reviewers’ comments and would recommend Accept.